# A Network Model for Identifying Key Causal Factors of Ship Collision

**Jianzhou Liu, Huaiwei Zhu, Chaoxu Yang and Tian Chai ***

College of Navigation, Jimei University, Xiamen 361021, China; 202112861003@jmu.edu.cn (J.L.);
zhw1645@163.com (H.Z.); y1810652154@163.com (C.Y.)
* Correspondence: chaitian@jmu.edu.cn

**Abstract:** In the analysis of the causes of ship collisions, the identification of key causal factors can help maritime authorities to provide targeted safety management solutions, which is of great significance to the prevention of ship collisions. In order to identify the key causal factors leading to ship collisions, we first construct a network model of ship collisions, in which the nodes represent the causal factors, and the edges represent the interrelationship between the causal factors. Second, based on the constructed network model, we propose a successive safety analysis method. This method can quantify the importance of each causal factor, and the quantified results allow us to identify the key causal factors of ship collisions. Finally, we verify the validity of the model using numerical cases.

**Keywords:** complex networks; cascading failures; causes of accidents; ship collisions

## 1. Introduction

Maritime transport is an important part of global trade, which accounts for 90% of the total global trade [1]. However, the maritime environment is complex and volatile, and it is extremely challenging to ensure the safe navigation of ships and avoid maritime traffic accidents. According to a related study [2], among the many maritime traffic accidents, ship collisions have the highest incidence rate. Meanwhile, the occurrence of ship collision accidents often causes serious consequences, such as casualties, property damage, and environmental pollution [3]. Therefore, how to prevent ship collision accidents has been the focus of attention of maritime departments.

Usually, accidents or unsafe events are caused by a number of uncertainties [4]. These uncertainties together form a complex system, and anomalies in multiple factors within the system may induce an accident to occur. In maritime traffic systems, ship collisions occur when multiple factors interact to induce the formation of causal chains that lead to accidents. In each causal chain, there are often some key factors, and the abnormality of the key factors will lead to the abnormality of a series of factors. Therefore, finding the key factors and managing them can reduce the chain reaction of the causal chain and thus reduce the probability of accidents.

In practical ship safety management, the identification of critical factors is very important. Due to the limited human and material resources on board, the crew cannot apply the same level of protection to all causes of accidents. Therefore, the identification of critical factors can help the crew to prioritize the needs of the causes of accidents. This significantly improves the efficiency of ship safety management. Based on this fact, we propose a network model to identify the critical causes of ship collisions. We analyze the model from the perspective of protection, quantify each cause of accident through network efficiency, and finally identify the critical causes of accidents through the quantified results.

The rest of the paper is structurally organized as follows: Section 2 describes the work related to ship collision accident studies. Section 3 describes the basic concept of cascading failures. Section 4 presents the construction method and identification method of the model.

Section 5 uses numerical cases to demonstrate the validity of the model. Section 6 presents the conclusions and future work.

## 2. Literature Review

At present, many scholars have conducted studies on ship collision accidents. According to their research directions, these studies can be divided into ship collision accident causation analysis and ship collision accident risk assessment.

The purpose of ship collision causation analysis is to explore the root causes of ship collisions and to reduce the probability of accidents by strengthening the protection against the root causes of accidents. For example, Zhang [5] used hierarchical analysis to establish a ship–bridge collision risk evaluation model, and this model can provide decision-making suggestions for bridge siting through the qualitative and quantitative assessment of the risk level of ship–bridge collisions. Ugurlu [6] used fault trees and multiple correspondence analyses to quantitatively and qualitatively analyze 513 ship collisions, and the final results of the study showed that 94.7% of the collisions were caused by human errors. Afenyo et al. [7] applied Bayesian networks to the collision scenario between ships and icebergs in order to study the root causes of ship–iceberg collisions in the Arctic shipping route, and through sensitivity analysis, they concluded that mechanical equipment, miscommunication, and communication equipment failure are the main causes of ship–iceberg collisions. Chai et al. [8] collected 300 reports of ship collisions, extracted the causal factors and causal chains from them, constructed a ship collision causal network with the causal factors as nodes, and analyzed the dynamic changes in the nodes of the network by setting different thresholds to finally determine the root causes of ship collisions.

Ship collision risk assessment aims to assess the collision risk of ships navigating in different water environments, and the results of their studies are usually the probability of ship collisions in different environments. For example, Zhen et al. [9] proposed a real-time multivessel collision assessment framework to evaluate the collision risk of ships in complex waters, in which the clustering method of spatial density was first used to identify ship encounter scenarios; then, the ship collision risk assessment function was constructed using the distance-closest point of approach (DCPA), the time-closest point of approach (TCPA) and relative bearing (RB), and finally, the collision risk of ships in complex waters was evaluated based on the magnitude of the function value. Wu et al. [10] proposed a fuzzy logic method for ship–bridge collision risk assessment, which takes ship characteristics, natural environment, and other factors as variables and fuzzifies these variables; after fuzzifying the input variables, IF–THEN rules are established and used for fuzzy reasoning to derive the bridge collision risk and determine the ship–bridge collision risk level. Huang et al. [11] used the speed method to evaluate the overlap probability of two ships' positions in the future by setting different reachable speeds and finally used the overlap probability to evaluate the collision risk of ships. Chen et al. [12] proposed a collision risk assessment method based on speed barriers, and they evaluated the collision risk between multiple ships from the perspective of speed and verified the effectiveness of this method by using the data of the automatic ship identification system.

There is no doubt that the above research studies have a positive impact on the prevention of ship collisions. However, with the rapid increase in the level of ship automation, and the potential causes of accidents have become more complex. In the process of analyzing the causes of accidents, it is necessary to consider the correlation between the causes of accidents as much as possible. However, the number of causes can be hundreds or thousands, and it will be a challenge to analyze the interrelationship of these causes effectively. Fortunately, complex networks can provide a good method for this challenge.

Complex networks originated in the 1980s and have been widely used in the field of safety engineering after many years of development. For example, traffic safety [13,14], construction safety [15,16], power security [17,18], and multimodal transport network security [19] have been studied. Meanwhile, a large number of studies have shown that complex networks can more clearly explain the correlation between things from a system

perspective [20]. Therefore, complex networks become an effective method to analyze the complex relationships between things. In particular, the use of complex networks has good application prospects in the field of traffic safety [21].

In this paper, we propose a network model to identify the key causal factors of ship collisions based on complex networks. We analyze the model from the perspective of protection and quantify each causal factor. The quantified results allow us to identify the key factors of ship collisions and provide theoretical guidance for ship safety management.

## 3. Cascading Failure Theory

A cascading failure is a phenomenon of anomalous information propagation in a network. In some real networks, a failure in one or a few nodes, followed by a failure in other nodes through the coupling between nodes, can cause successive failures in other nodes, which can create a chain reaction that eventually leads to the collapse of a significant portion or even the entire network [22]. There are many models for analyzing cascade failures in complex networks in which each node is assigned an initial load and load capacity, and when the load of a node exceeds its capacity, that node will fail, and its load will be distributed to its neighboring nodes. At this point, the initial load of the neighboring node will also change as the neighboring node accepts the additional load. Consequently, if the load of the neighbor node exceeds its capacity, a new failed node will appear, which in turn will lead to a new round of load distribution.

The load capacity model is one of the common models of cascading failures [23]. Generally, the load capacity model assigns the initial load and capacity to each node in the network and determines whether the fault spreads by comparing the load value and capacity value of the node. According to the basic concept of cascading failures [24], the initial node load, node capacity, and some other parameters are defined as follows:

(1)    Initial load

In 2003, Motter et al. [25] proposed a load capacity model in which the information and energy of nodes in a network are propagated according to the shortest paths between pairs of nodes. Therefore, the initial load size of a node can be expressed in terms of the number of shortest paths through the node. In a complex network, the number of shortest paths through a node can be expressed in terms of the betweenness. Therefore, in this paper, the initial load of a node is expressed by its betweenness, and the formula for calculating the betweenness of a node is shown in Equation (1).

$$D_i(0) = \sum_{i,j \in V(j \neq k)} \frac{n_{jk}(i)}{n_{jk}} \tag{1}$$

where $n_{jk}$ represents the number of shortest paths connecting points $j$ and $k$; $n_{jk}(i)$ represents the number of shortest paths connecting points $j$ and $k$ and passing through point $i$.

(2)    Node capacity

In the load capacity model, node capacity refers to the maximum load value that a node can carry. Node capacity is the threshold value that determines fault propagation, and when the node load is greater than its capacity, the node will fail and propagate the load to adjacent nodes. Node capacity is generally proportional to its initial load, and its calculation formula is shown in Equation (2).

$$C_i = (1 + \lambda)D_i(0) \tag{2}$$

where $\lambda$ is the tolerance factor.

## 4. Method

### 4.1. Ship Collision Causation Network Structure

The key step in constructing the causal network model of ship collision is to determine the nodes in the network and the rules of the connected edges between the nodes. In a real network, the objects of study are generally abstracted as nodes, and the interrelationships between the objects of study are used as edge rules. In the same way, we abstracted the causal factors of ship collision as network nodes and the interrelationship between the causal factors as the connected edges between nodes. The specific network construction steps were as follows:

Step 1: Collect reports of ship collisions;
Step 2: Analyze the ship collision report and extract the causal factors from it;
Step 3: Count the causal factors contributing to the same accident;
Step 4: Construct a causal network of ship collision accidents. The causal factors were used as network nodes, and edges were defined as the interre-lationship between the causal factors that appeared in the same acci-dent.

### 4.2. Successive Security Evolutionary Processes

Based on the constructed network model and according to the idea of cascading failures, we propose a successive safety evolution process for ship collisions. In this study, the load value of each node in the network model is defined as the safety protection strength of the causative factor, the initial load of the node is considered as the initial protection value, and the capacity of the node is defined as the safety protection threshold. If the safety protection strength of a node is greater than the safety threshold of that node, we consider the node as a safe node, i.e., the node will not cause an accident, and thus it is removed from the network. However, due to the complexity and variability of the ship's navigation environment and the limitation of human and material resources in ship safety management, sufficient protection strength cannot be given to each causative factor. As a result, the initial protection value of each node is often lower than its safety threshold.

In this paper, the concept of successive safety is proposed based on the theory of cascading failures. Therefore, the initial load of the node is the initial protection value of the node, and the calculation formula is shown in Equation (3).

$$P_i(t) = \sum_{i,j \in V(j \neq k)} \frac{n_{jk}(i)}{n_{jk}} \tag{3}$$

The safety protection threshold for the node is $C_i = (1 + \lambda)P_i(0)$.

When the protection value of node $i$ exceeds the corresponding safety threshold, the node will be in a secure state, at which time any additional protection beyond the safety threshold of this node will be equally divided among its neighbors $j$, calculated as follows:

$$\Delta P_{ij} = \frac{P_i(t) - C_i}{d_i(t)} (j \in \Gamma_i) \tag{4}$$

where $\Delta P_{ij}$ denotes the protection value passed from node $i$ in a secure state to neighbor node $j$; $P_i(t)$ denotes the protection value of node $i$ at moment $t$; $d_i(t)$ denotes the number of neighbor nodes of node $i$ at moment $t$; $C_i$ denotes the node safety threshold; $\Gamma_i$ denotes the set of neighbor nodes of node $i$.

After calculation, the neighboring nodes of node $i$ obtain additional protection values, and therefore their own protection values will change as follows:

$$\begin{cases} P_j(t) = P_j(t-1) + \Delta P_{ij} = P_j(t-1) + \frac{P_i(t-1) - C_i}{d_i} \\ P_j(t+1) = \begin{cases} P_j(t) & (P_j(t) \leq C_i) \\ C_i & (P_j(t) > C_i) \end{cases} \end{cases} \tag{5}$$

The evolution process of "successive safety" of the causal network model of ship collision proposed in this paper is shown in Figure 1. The number in the circle in Figure 1 indicates the protection rate of the node, i.e., protection rate = protection value of the node/safety threshold of the node × 100%. When the protection rate reaches 100%, the node is in a safe state, and the node will not cause an accident. When the protection rate exceeds 100%, the node maintains a safe state while assigning the additional protection value above the safety threshold to its neighboring nodes. For example, when $t = 0$, the protection rate of each node in the network does not reach 100%; at this time, each node is in a dangerous state, and these nodes may cause an accident at any time. When $t = 1$, additional protection measures are applied to node 4, so that the protection value of node 4 exceeds its safety threshold. At this point, the protection rate of node 4 exceeds 100%, so node 4 has additional protection values to assign to its neighboring nodes (such as node 5 and node 6). At $t = 2$, the protection values of node 5 and node 6 are updated as node 5 and node 6 receive additional protection values, and the protection rates of both node 5 and node 6 exceed 100% after the update. At this time, node 5 and node 6 have additional protection values to assign to their neighboring nodes (e.g., node 1 and node 7). At t = 3, after node 1 and node 7 receive additional protection values, their respective protection values are updated, and the protection rate of node 7 exceeds 100% after the update. At this point, node 7 has additional protection values to distribute to its neighboring nodes (e.g., node 3). At $t = 4$, node 3 receives the additional load, and its protection rate exceeds 100%. At this point, node 3 allocates the additional protection value to node 2. At $t = 5$, the "successive safety process" in the network ends because no new node has a protection rate exceeding 100%.

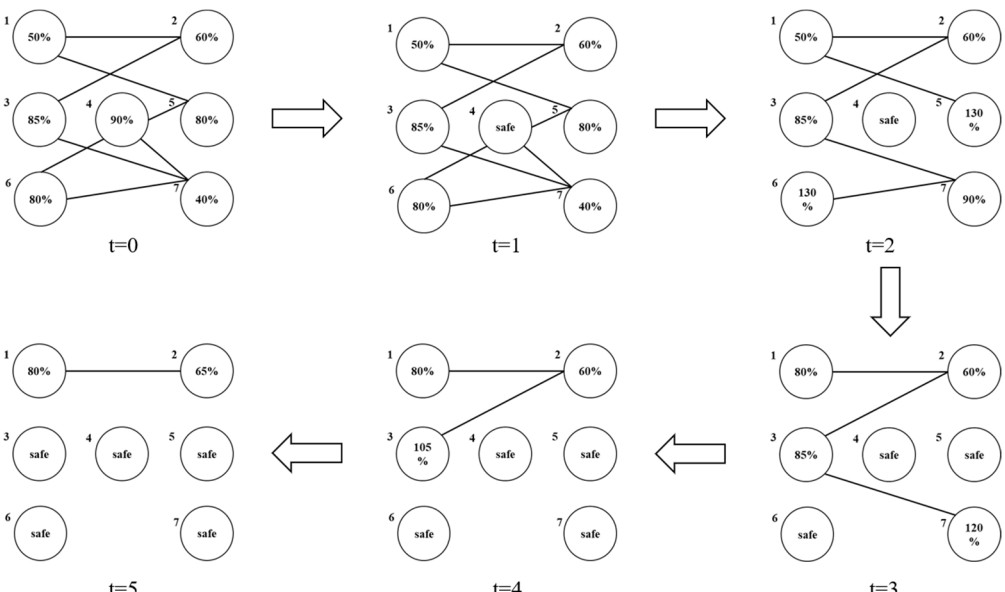

**Figure 1.** Successive safety evolution process.

### 4.3. Network Efficiency

According to the successive safety process, the initial node entering the secure state triggers the successive safety evolution process of other nodes in the network. Therefore, in order to evaluate the degree of impact on the network as a whole after the initial node triggers the successive safety process, we introduce the evaluation index of network efficiency. The network efficiency is calculated as follows:

$$E(i) = \frac{N'(i)}{N} \tag{6}$$

where $N'(i)$ is the number of nodes remaining in the network after successive failures of node i, and $N$ is the initial number of nodes in the network.

## 5. Numerical Case Study

### 5.1. Constructing a Causal Network for Ship Collisions

In this study, we collected 300 reports of ship collisions that occurred in Chinese waters during a 20-year period from 1999 to 2018. From the reports, we extracted 98 causal factors (see Table A1). According to the network model construction method in Section 4.1, we successfully constructed a causal network of ship collisions, as shown in Figure 2.

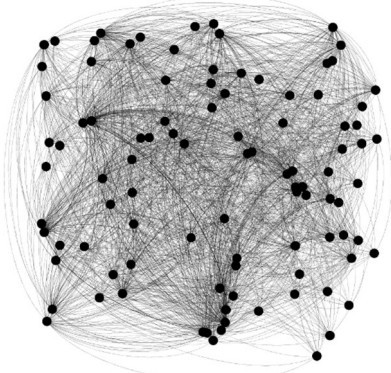

**Figure 2.** Ship collision causation network model.

### 5.2. Successive Safety-Triggering Processes

In complex networks, the node with a large degree value is called a hub node, which is coupled with more nodes in the network. In a ship collision causation network, the hub node represents the factor that causes more accidents, and more human and material resources need to be invested in the protection of this type of node to prevent its failure. Therefore, the hub node has a higher initial protection value and safety threshold than other nodes. By analyzing the successive safety processes triggered by hub nodes, it is possible to recognize the important role played by hub nodes in the network, which is of great significance for accident prevention.

The degree values and initial protection values of each node in the ship collision causation network model were calculated, and the results are shown in Figures 3 and 4.

From Figures 3 and 4, it can be seen that the degree value and initial protection value of node 1 (improper lookout) are the largest, so node 1 is used as the hub node in the network. Next, we take this node as an example to analyze the successive safety process of this node in the network.

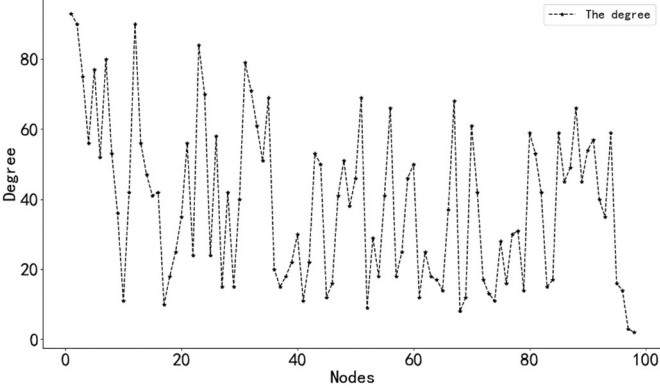

**Figure 3.** The degree value of each node.

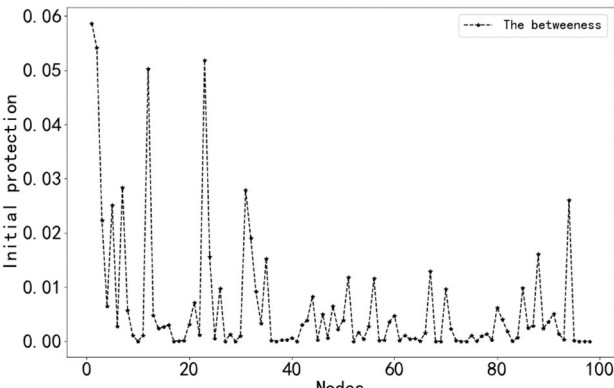

**Figure 4.** The initial protection value of each node.

Before analyzing the successive safety process of the hub node, we briefly highlight the following assumptions:

Assumption 1: The tolerance factor is set to 0.5, which means that the protection of the causal factor reaches 1.5 times its safety threshold to ensure that the causal factor is in a safe state, i.e., the node will not cause a ship collision to occur.

Assumption 2: In order to trigger the successive safety propagation process, additional protection values need to be assigned to the hub node. In this paper, the hub node is assigned an additional protection value equal to three times its own protection value, so that it is in a fully safe state. At this point, the protection value of the hub node is greater than its own safety threshold, so it will promote the successive safety process.

Assumption 3: When a node's protection value exceeds its safety threshold, the part of the node's protection value that exceeds the safety threshold is assigned to a neighboring node, and the node is removed from the network. When the protection values of all nodes in the network are not affected by other nodes, the process of successive safety propagation ends. At this point, the protection values of all nodes in the network are less than their corresponding safety thresholds.

Based on the above assumptions, we take the hub node as an example to analyze the successive safety evolution process of the causative network of ship collisions, and the specific evolution process is explained in what follows.

t = 0: As the initial protection value of node 1 does not reach the safety threshold, node 1 is assigned an additional protection value equal to three times its own protection value to make it in a fully secure state. At this point, the protection value of node 1 is greater than its own safety threshold, and the successive safety evolution starts.

t = 1: The protection values of neighboring nodes are updated through the successive safety evolution of the protection value of node 1. The updated protection values of neighboring nodes are compared with their own safety thresholds, as shown in Figure 5.

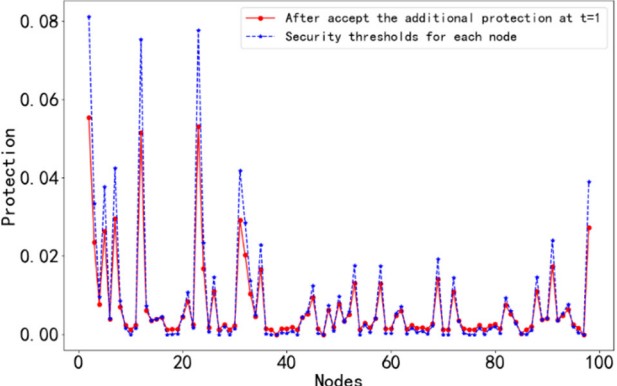

**Figure 5.** The protection value and safety threshold of each node at t = 1.

From Figure 5, the nodes that enter a fully secure state in the network after passing the first successive safety evolution were determined, and the results are shown in Table 1.

**Table 1.** Node numbers in a fully secured state.

| Node Numbers in Fully Secured State | | | | | | | | | |
|---|---|---|---|---|---|---|---|---|---|
| 8 | 9 | 10 | 13 | 16 | 17 | 18 | 21 | 24 | 26 |
| 48 | 50 | 53 | 54 | 55 | 58 | 59 | 62 | 63 | 64 |
| 65 | 66 | 67 | 69 | 70 | 72 | 73 | 74 | 75 | 76 |
| 77 | 78 | 79 | 80 | 83 | 85 | 86 | 88 | 91 | 94 |
| 95 | | | | | | | | | |

t = 2: Repeating the above successive safety evolution process, the protection value of neighboring nodes at the moment t = 2 is compared with its own safety threshold, as shown in Figure 6.

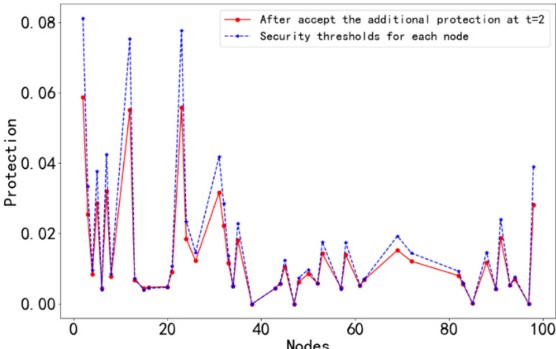

**Figure 6.** The protection value and safety threshold of each node at t = 2.

The nodes that enter a fully secure state in the network after passing the second successive safety evolution were derived from Figure 6, and they are shown in Table 2.

**Table 2.** Node numbers in a fully secured state.

| Node Numbers in Fully Secured State | | | | |
|---|---|---|---|---|
| 5 | 14 | 15 | 19 | 33 |
| 51 | 56 | 89 | 92 | 96 |

t = 3: Similarly, the above successive safety evolution process is repeated to obtain a comparison of the safety threshold of the neighboring node protection with its own at the moment t = 3, as shown in Figure 7.

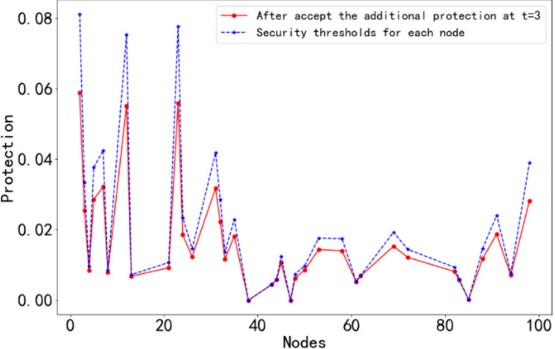

**Figure 7.** The protection value and safety threshold of each node at t = 3.

As can be seen from Figure 7, only node 84 is in a fully secure state after the third successive safety evolution.

t = 4: Continuing the above process of successive safety evolution, a comparison of the t = 4 neighbor node protection with its own safety threshold is obtained, as shown in Figure 8.

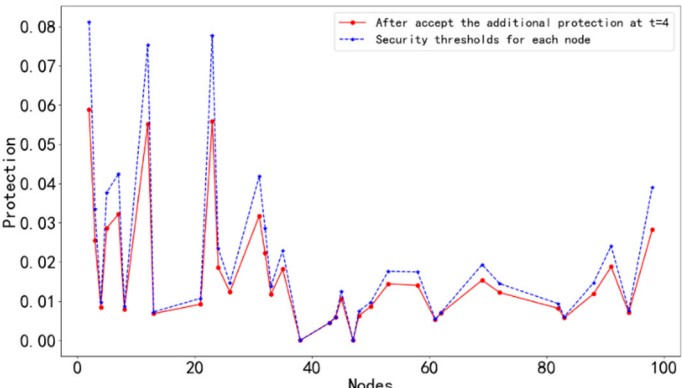

**Figure 8.** The protection value and safety threshold of each node at t = 4.

As can be seen from Figure 8, at this time, the protection values of the remaining nodes in the network are less than their own safety threshold, and the successive safety evolution process ends.

Throughout this process, the successive evolution toward a secure state triggered by node 1 leads to some nodes entering the secure state at each step in the network. In order to more intuitively reflect the changes in the nodes of the network, we plotted the changes in the number of nodes at each step of the successive safety evolution process, and the node changes are shown in Figure 9.

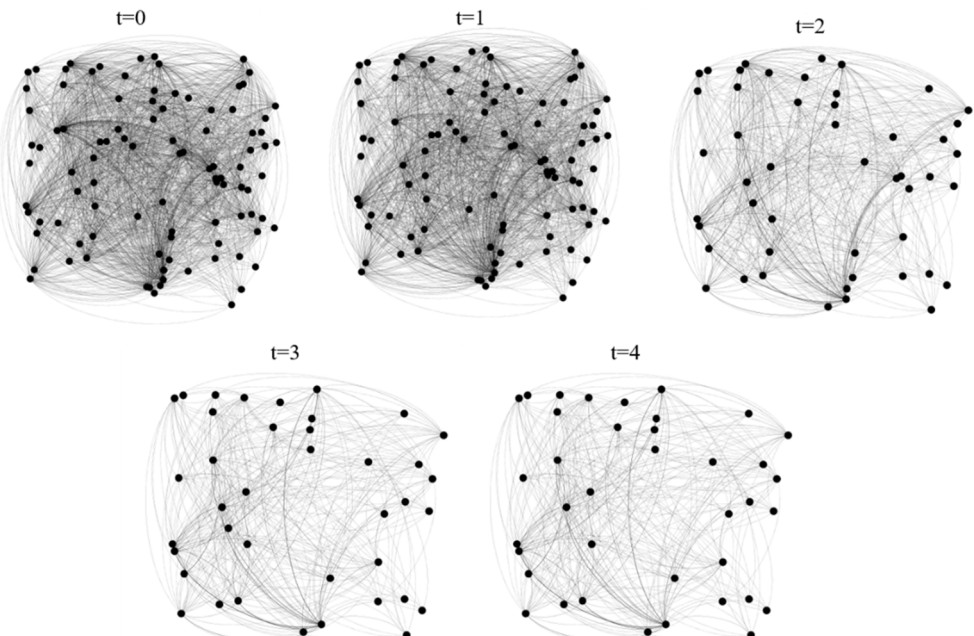

**Figure 9.** The evolution of the causal network of ship collisions.

As can be seen from Figure 9, the network gradually becomes sparse after each evolutionary step. This means that the successive safety processes triggered by node 1 have more causative nodes entering a secure state. The reduction in the causative nodes in the network can directly indicate that the protection of node 1 can reduce the probability of accidents.

According to Section 4.3, we calculated the network efficiency values after each successive safety evolution, and the calculation results are shown in Table 3. It can be seen that as the successive safety evolution process continues, the network efficiency decreases. This indicates that there are fewer and fewer factors in the causal network that can induce ship collisions, and the probability of ship collisions is increasingly minimized.

**Table 3.** Network efficiency changes.

| Time | t = 0 | t = 1 | t = 2 | t = 3 | t = 4 |
|------|-------|-------|-------|-------|-------|
| N′ | 98 | 97 | 46 | 36 | 35 |
| E(1) | 1.0000 | 0.9899 | 0.4694 | 0.3673 | 0.3571 |

In summary, the analysis of the successive safety evolution process triggered by node 1 shows that the hub node ends after four successive safety evolution processes. The causative nodes in the network are reduced from 98 to 35, and the network efficiency is reduced from 1.0000 to 0.3571. This indicates that the protection of node 1 can greatly reduce the incidence of accidents.

### 5.3. The Successive Safety Evolution Process of Each Node

In order to evaluate the importance of each node in the network, we performed the above steps for each node so that each node triggered the successive safety process. Finally, the network efficiency of each node at the end of the successive safety process was obtained (see Table 4).

**Table 4.** Network efficiency value of each node.

| ID | Network Efficiency | ID | Network Efficiency | ID | Network Efficiency | ID | Network Efficiency |
|----|--------------------|----|--------------------|----|--------------------|----|--------------------|
| 1 | 0.3571 | 26 | 0.8775 | 51 | 0.9591 | 76 | 1.0000 |
| 2 | 0.3775 | 27 | 1.0000 | 52 | 0.9489 | 77 | 0.9591 |
| 3 | 0.6734 | 28 | 0.9795 | 53 | 0.8367 | 78 | 0.9897 |
| 4 | 0.9081 | 29 | 1.0000 | 54 | 1.0000 | 79 | 0.9897 |
| 5 | 0.5918 | 30 | 0.9897 | 55 | 0.9591 | 80 | 0.9693 |
| 6 | 0.9693 | 31 | 0.6122 | 56 | 0.9693 | 81 | 0.9795 |
| 7 | 0.6020 | 32 | 0.7448 | 57 | 0.9489 | 82 | 0.9489 |
| 8 | 0.9183 | 33 | 0.8775 | 58 | 0.8265 | 83 | 0.9693 |
| 9 | 0.9897 | 34 | 0.9387 | 59 | 0.9897 | 84 | 0.9795 |
| 10 | 1.0000 | 35 | 0.7857 | 60 | 0.9897 | 85 | 0.9795 |
| 11 | 0.9897 | 36 | 0.9897 | 61 | 0.9489 | 86 | 0.9897 |
| 12 | 0.4183 | 37 | 1.0000 | 62 | 0.9285 | 87 | 0.9591 |
| 13 | 0.9183 | 38 | 1.0000 | 63 | 0.9897 | 88 | 0.9081 |
| 14 | 0.9897 | 39 | 0.9795 | 64 | 0.9693 | 89 | 0.9897 |
| 15 | 0.9489 | 40 | 0.9795 | 65 | 0.9897 | 90 | 0.9693 |
| 16 | 0.9489 | 41 | 0.9897 | 66 | 0.9795 | 91 | 0.8061 |
| 17 | 1.0000 | 42 | 1.0000 | 67 | 0.9897 | 92 | 0.9693 |
| 18 | 0.9897 | 43 | 0.9693 | 68 | 0.9897 | 93 | 0.9693 |
| 19 | 0.9897 | 44 | 0.9285 | 69 | 0.8571 | 94 | 0.9285 |
| 20 | 0.9693 | 45 | 0.9081 | 70 | 1.0000 | 95 | 0.9795 |
| 21 | 0.9387 | 46 | 0.9897 | 71 | 1.0000 | 96 | 0.9897 |
| 22 | 0.9489 | 47 | 1.0000 | 72 | 0.8265 | 97 | 1.0000 |
| 23 | 0.3979 | 48 | 0.9285 | 73 | 0.9693 | 98 | 0.8061 |
| 24 | 0.8265 | 49 | 0.9897 | 74 | 0.9795 | | |
| 25 | 0.9795 | 50 | 0.8979 | 75 | 1.0000 | | |

In order to analyze the above results more clearly, we drew a network efficiency diagram at the end of the successive safety processes of each node, as shown in Figure 10. At the same

time, the probability of each causative factor triggering the occurrence of 300 accidents was counted, and the probability statistics are shown in Figure 11.

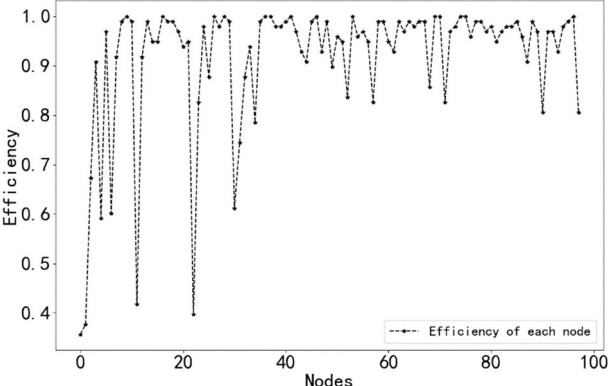

**Figure 10.** The network efficiency of each node.

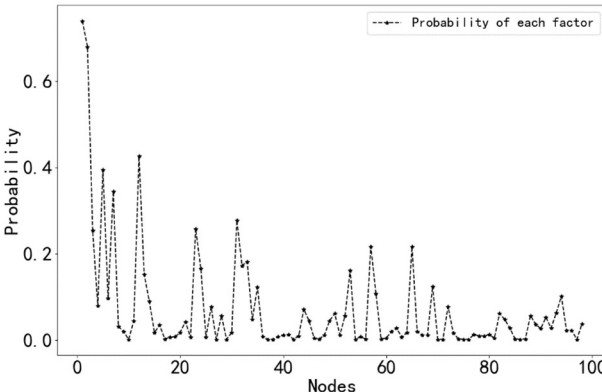

**Figure 11.** The probability of induced accidents at each node.

In the existing studies, the number of accidents caused is mostly used directly as an important indicator to evaluate the degree of influence of causal factors on ship collisions. Here, we used network efficiency to measure the importance of accident causation. The comparison shows that the evaluation results of these two indicators on the importance of accident causation are basically the same, but there are also some differences. For example, the probability of accidents caused by node 22 is smaller than that of nodes 21 and 23, but the reduction in network efficiency caused by the control of node 22 is much higher than that of nodes 21 and 23, which means that although the number of ship collision accidents caused by the causal factor represented by node 22 is smaller, the incidence of ship collision accidents can be effectively reduced by investing human and material resources in its prevention and control, so the degree of influence of this factor on the incidence of ship collision accidents is much higher than that of nodes 21 and 23. Therefore, the influence of this factor on the occurrence rate of ship collisions is also greater. Secondly, the objects of these two evaluation indexes are different. The calculation of accident probability is influenced by the number of collected accidents, which is more contingent and not general. On the other hand, the network efficiency is determined from the perspective of the protection of causal factors and enables the analysis of the influence of causal factors on the occurrence rate of ship collision accidents under the same level of protection, which is objective and not influenced by the number of collected accidents. With the improvement in ship intelligence, the human and material resources invested in ship safety management are more valuable, and the reasonable allocation of human and material resources in ship safety management has an important influence on the reduction in ship collision rates. Thus, the method proposed in this paper can provide a valuable reference for the reasonable allocation of human and material resources.

In addition, as new routes are opened (e.g., Arctic routes), new causal factors of ship accidents will emerge. However, the ship collision causation network designed in this paper is an open network, which is updateable. For new causes of accidents, we can update the network at any time according to the construction steps of the network described in Section 4.1. By analyzing the new network, we can assess the influence of new causal factors on accident occurrence. This is more in line with the realistic needs resulting from variability in the ship navigation environment.

## 6. Conclusions

In this paper, a network model of ship collision accidents was established based on the complex network theory. In order to evaluate the importance of causative nodes in the network model, we proposed a successive safety analysis method. The concepts of the initial protection value, the safety threshold, and the protection rate of each node were introduced to help us control the spread of ship collisions by triggering a successive safety evolution process, and each causative factor was quantified according to network efficiency. Lastly, the key causative factors of ship collisions were identified based on the quantified results.

Numerical case studies show that improper lookout is the key cause of accidents, and the probability of accidents will be reduced to less than 40% by taking protective measures against this cause of accidents. Therefore, in ship navigation, the watchkeeping officershould consciously abide by the terms of lookout procedures, use a combination of visual and radar observations for lookout, strictly follow policies while on duty, and refrain from any behavior that affects the driver's formal lookout, such as drunk driving and fatigue driving.

Finally, compared with most of the other existing methods, our proposed method concerns accident protection and enables the quantification of individual causal factors of accidents, and the quantified results are more generalized regardless of the number of collected accidents. In addition, the network model we constructed is an open model, and we will apply this model to special navigation environments (e.g., Arctic routes) in future research to enrich the database of the model and further expand its application scenarios.

**Author Contributions:** Conceptualization, J.L.; methodology, J.L.; software, J.L.; validation, J.L.; formal analysis, J.L.; investigation, H.Z.; resources, H.Z.; data curation, T.C.; writing—original draft preparation, J.L.; writing—review and editing, J.L. and T.C.; visualization, C.Y.; supervision, T.C.; project administration, T.C. All authors have read and agreed to the published version of the manuscript.

**Funding:** This research is funded by the Natural Science Foundation of Fujian Province (Grant No. 2019J01326).

**Institutional Review Board Statement:** Not applicable.

**Informed Consent Statement:** Not applicable.

**Data Availability Statement:** Not applicable.

**Conflicts of Interest:** The authors declare no conflict of interest.

## Appendix A

**Table A1.** The causal factors of ship collisions.

| Cause ID | Cause Name |
|---|---|
| 1 | Improper lookout |
| 2 | Inappropriate assessment of the situation of the risk of collision |
| 3 | Did not make a sound signal as per guidelines |
| 4 | Did not exhibit lights and shapes as per guidelines |
| 5 | Did not navigate at a safe speed |

**Table A1.** *Cont.*

| Cause ID | Cause Name |
| --- | --- |
| 6 | Did not take actions such as slacking a vessel's speed, stopping or reversing her propulsion in ample time to avoid close quarters |
| 7 | Did not take actions such as slacking a vessel's speed, stopping or reversing her propulsion in ample time to avoid close quarters |
| 8 | The person on duty was not on the bridge |
| 9 | VHF was not on duty |
| 10 | Misunderstood the information from the VHF |
| 11 | Took action blindly |
| 12 | Did not take effective action in good time to avoid collision (miss the best time to take effective action) |
| 13 | The give-way vessel did not carry out the duty to keep out of the way |
| 14 | The stand-on vessel took action in error |
| 15 | The stand-on vessel took action in error |
| 16 | Deviation from specified course |
| 17 | Violation of the regulation of no-drinking when on duty |
| 18 | Dozed off on duty |
| 19 | The person on duty was engaged in something irrelevant to navigation |
| 20 | Did not keep a safe distance from the anchoring ship |
| 21 | Did not obtain sufficient information about the surrounding navigation environment |
| 22 | Did not monitor own ship's position sufficiently |
| 23 | Did not meet the requirements of good seamanship and seafarers' usual practice |
| 24 | Inadequate use of radar/ARPA |
| 25 | Identified the information of radar target in error |
| 26 | Inadequate use of AIS (including not installing or turning on AIS) |
| 27 | The captain did not give any orders for night navigation |
| 28 | The captain failed to command regarding the situation of the bridge as required (e.g., foggy weather, narrow water channel, and traffic-dense area) |
| 29 | The navigation alarm on the bridge was not on |
| 30 | The steering device was not used at the proper time |
| 31 | Did not see other ships as early as possible |
| 32 | Failure to track or misjudge the dynamics of other ships |
| 33 | Did not take coordinated turning actions in an urgent situation |
| 34 | Took improper emergency measures to avoid collision |
| 35 | Did not take the most helpful actions to avoid collision in an emergency risk |
| 36 | Overtook blindly without other ships' approval |
| 37 | Communication failure between the officer on duty and the sailor on duty |
| 38 | Communication error between the bridge control and the engine room |
| 39 | The officer is not familiar with the rules of the COLREGs |
| 40 | The officers' shift error |
| 41 | The inexperience of the person on duty |
| 42 | The person is sitting when he is on duty |
| 43 | The officer is not familiar with the maneuverability of the ship |
| 44 | Small alterations in course to avoid collision |
| 45 | Underestimated the impact of wind, wave, and current |
| 46 | No tug assistance was applied when berthing or unberthing (no application of tug assisting when berthing and unberthing) |
| 47 | Operated the tug improperly when berthing and unberthing |
| 48 | The remaining speed was too fast when berthing and unberthing |
| 49 | Did not check the effectiveness of the action to avoid collision |
| 50 | Violation of navigation regulations of the water area (including regulations on ship routing system) |
| 51 | Not following VTS advice or traffic control |
| 52 | Failure to comply with narrow channel navigation rules |
| 53 | The incompetence of the crew member |
| 54 | The officer's overfatigue |
| 55 | Pilot operation error |
| 56 | The officer left the ship too early |
| 57 | Failure to obey report obligation |
| 58 | Insufficient crew number |

**Table A1.** *Cont.*

| Cause ID | Cause Name |
| --- | --- |
| 59 | Machine failure without repair guarantee |
| 60 | Ships sailing beyond the approved navigation area |
| 61 | Anchoring in waterway, customary route, or dense-traffic area |
| 62 | Violation of mooring duty requirements |
| 63 | Improper anchoring method |
| 64 | No effective monitoring of anchorage position during anchoring |
| 65 | No effective measures were taken after anchor dragging |
| 66 | Significantly affected by the wind |
| 67 | Significantly affected by the wave |
| 68 | Significantly affected by the current |
| 69 | Poor visibility |
| 70 | The effect of navigational obstructions |
| 71 | The effect of the bend of the channel |
| 72 | Navigation impact in traffic-dense areas (complicated navigation environment) |
| 73 | The impact of collision avoidance by third-party vessels |
| 74 | The impact of narrow waterways |
| 75 | The impact of shallow water |
| 76 | The VHF communication channel was too noisy |
| 77 | The main engine broke down |
| 78 | Failure of the steering gear |
| 79 | AIS fault |
| 80 | Other facilities' failure |
| 81 | The influence of the blind area of the bow |
| 82 | The ship was not in a seaworthy condition |
| 83 | The Vessel certificate expired or undocumented |
| 84 | No ship inspection was conducted as required |
| 85 | The ship failed to correct safety defects or faults before sailing |
| 86 | No VTS was established |
| 87 | VTS supervision error |
| 88 | The shipping company failed to fulfill the main responsibility of safety production |
| 89 | The shipping company commanded the ship to operate on sea illegally |
| 90 | The shipping company did not provide enough qualified crew for the ship |
| 91 | The shipping company did not establish SMS or the requirements of SMS were not implemented |
| 92 | The training and assessment of the crew members by the shipping company were insufficient |
| 93 | (Improper arrangement of persons on duty) insufficient staff on duty |
| 94 | No additional lookout staff |
| 95 | The shipowner did not sufficiently know about the information and competency of the crewmembers |
| 96 | The shipping company did not monitor the ship dynamically |
| 97 | The shipping company did not fully grasp the navigation management regulations important to ship safety |
| 98 | Defects in bridge resource management |

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
