# Peer review of "A Network Model for Identifying Key Causal Factors of Ship Collision"

_jmse, doi:10.3390/jmse11050982_

Round 1
Reviewer 1 Report
The authors are very familiar with the fields. The contributions of the work are:
- The authors established a network model of ship collisions based on complex network theory with the proposal of a method of successive safety analyses.
- The paper has an excellent structure with a clear explanation.
- The steps are well explained.
- The developed methodology is given in detail.
- Good problem definition.
The paper has great potential and can be accepted after the following MINOR corrections:
1. Line 77: During the first indication of the turning points, it is necessary to indicate the full name and in parentheses indicate the abbreviation that will be used in further work (for example: Relative bearing (RB))
2. In order to improve the quality, it is necessary to look at recent works that deal with the issue of theory of cascade failures: for example
2.1. H. Guo, S.S. Yu, H. H. C. Iu, T. Fernando, C. Zheng, A complex network theory analytical approach to power system cascading failure From a cyber-physical perspective, Chaos 29, 053111 (2019), https://doi.org/10.1063/1.5092629
2.2. Z. He, J.N. Guo, J.X. Xu, Cascade Failure Model in Multimodal Transport Network Risk Propagation, Volume 2019, Article ID 3615903, https://doi.org/10.1155/2019/3615903
2.3. X. Deng, Sh. Wang, W. Wang, P. Yu, X. Xiong, 2023, Optimal defense strategy for AC/DC hybrid power grid cascading failures based on game theory and deep reinforcement learning, Frontiers in Energy Research 11, DOI: 10.3389/fenrg.2023.1167316
3.In accordance with the guidelines, the following is an addition to the manuscript review, jmse-2373035:
3.1 The paper deals with the area of a large number of conditionally related facts (causes of ship collisions). A large number of possible examples of ship collisions sets the task of looking at possible ways of solving problems in order to reduce the probability of accidents by strengthening protection against the basic causes of accidents.
3.2 Explain why the concept of successive safety is proposed based on the theory of cascading failures.
3.3 Explain whether the flow direction can be influenced by increasing the capacity of the node? For example, defining a "directed critical path"
3.4 The conclusion should be supplemented with a proposal for solving the problem of improper supervision. Would the introduction of certain defined SOPs or SETs (if there are more SOPs) be possible solutions?
Reviewer 2 Report
One of the serious problems of modern seafaring navigation is occurring of accidents. One of the most severe and numerous types of accidents is collisions. Statistics show that for the last decades the number of collisions has not significantly decreased in spite of extensive equipping of ships with modern means of monitoring, automatic information processing, communication and introduction of recommended ocean routes, traffic separation systems and other measures. Therefore, ensuring safe navigation by preventing ship collisions is an urgent task.
In order to solve this problem efficiently, it is necessary to take into account possible potential causes of accidents. However, their number is constantly growing. At the same time, cause-and-effect relations between them become more complicated. Therefore, to improve shipping safety, the authors suggest creating a network model to determine the key causal factors of ship collisions. In this case it is proposed to use methods of construction of complex networks. The article presents an algorithm for building such a network. Causal factors of an accident act as nodes in the network. If several factors appear in the same incident, a link (edge) is created between the nodes.
After creating a complex network model, the authors proceed to the sequential iterative process of improving the safety of navigation. For this purpose, they propose to use the method of cascade failure theory, the application of which makes it possible to reduce the size of the causal network of incidents. In the article, the authors described in detail the concept of sequential safety based on the cascade failure theory.
To assess the effectiveness of the proposed methods to reduce the probability of ship collisions, the authors conducted research using accident statistics in Chinese waters from 1999 to 2018. This allowed to identify 98 causal factors of ship collisions. Based on this data, a causal network of ship collisions was constructed. Complex network construction methods were used. After identifying the node that is the factor causing the greatest number of accidents, the authors proceed to a sequential iterative process to improve navigational safety. For this purpose, at each stage, the nodes are identified that move to a safe state. These nodes (factors) are excluded from the network. As a result, there is a reduction in the size of the causal network of accidents from 98 to 35. As a result of the protective measures, the probability of accidents in maritime transport has been reduced by 36%. Thus, the goal of the article has been achieved.
The scientific significance of the article lies in the fact that the authors have developed the concept of sequential safety by integrating the methods of complex network theory and the methods of cascade failure theory. The application of cascading failure theory in this concept made it possible to reduce the dimensionality of a complex network and to identify the key causal factors of ship collisions. As a result, the efficiency of safe management of ship traffic will be improved.
Practical significance of the work is that the proposed method of determining the key causal factors of ship collisions can be used by specialists responsible for the safety of marine vessel traffic. The developed network model can be supplemented with new factors influencing the collision of ships. That fact indicates its potential and promising use.
Remarks.
1. There is a typo in the formula on line 177. Instead of D(0) the authors should write Р(0).
English language and style is ok
